# No-regret Learning in Repeated First-Price Auctions with Budget Constraints

## Abstract

Recently the online advertising market has exhibited a gradual shift from second-price auctions to first-price auctions. Although there has been a line of works concerning online bidding strategies in first-price auctions, it still remains open how to handle budget constraints in the problem. In the present paper, we initiate the study for a buyer with a budget to learn her online bidding strategies in repeated first-price auctions. We propose an RL-based bidding algorithm against the optimal non-anticipating strategy under stationary competition. Our algorithm obtains $\widetilde{O}(\sqrt{T})$-regret if the bids are all revealed at the end of each round, where $\widetilde{O}(\cdot)$ is a variant of the big-$O$ that hides logarithmic factors. With the restriction that the buyer only sees the winning bid after each round, our modified algorithm obtains $\widetilde{O}(T^{\frac{7}{12}})$-regret by techniques developed from survival analysis. Our analysis extends to the more general scenario where the buyer can have any bounded instantaneous utility function with regrets of the same order. Simulation experiments show that the constant factor inside the regret bound is rather small.

## 1 Introduction

There has been extensive growth in the online advertising market in recent years. It was estimated that the volume of online advertising worldwide would reach 500 billion dollars in 2022 (Statista, 2021). In such a market, advertising platforms use auctions to allocate ad opportunities. Typically, each advertiser has a limited amount of capital for an advertisement campaign. Therefore, consecutive rounds of competition are *interconnected by budgets of participating advertisers*. Furthermore, each advertiser has very limited knowledge of 1) her valuation of certain keywords and 2) the competitors she is facing. There are many works being devoted to studying algorithms for *learning* strategies for optimally spending the budget in repeated *second*-price auctions (see Section 1.1).

In practice, on the other hand, we have witnessed numerous switches from second-price auctions to first-price auctions in the online advertising market. A recent remarkable example is Google AdSenses' integrated move at the end of 2021 (LLC, 2021). Earlier examples also include AppNexus, Index Exchange, and OpenX (Sluis, 2017). This industry-wide shift is due to various factors including a fairer transactional process and increased transparency. Therefore, the shift to first-price auctions brings about major importance to the following open question which is barely considered in previous works:

> *How should budget-constrained advertisers learn to compete in repeated* first-price *auctions?*

This paper thus initiates the study of learning to bid with budget constraints in repeated first-price auctions. It has been noted that the application of first-price auctions with budgets is not limited to online advertising mentioned above. Traditional competitive environments like mussel trade in Netherlands (van Schaik et al., 2001), modern price competition, and procurement auctions (e.g. U.S. Treasury Securities auction (Chari and Weber, 1992)) are examples as well.

**Challenges and contributions**    The challenges in this setting are two-fold.

The first challenge relates to the specific information structure of first-price auctions. In practice, it is often the case that only the highest bid is revealed to all participants (Esponda, 2008). This is known as *censored*-feedback or an informational version of *winner's curse* in literature (Capen et al., 1971).

This affects the information structure of learning since the buyer learns less information when she wins. This makes the problem challenging compared to standard contextual bandits (c.f. Section 1.1).

The second challenge is more fundamental. It is known that the strategy in first-price auctions is notoriously complex to analyze, even in the static case (Lebrun, 1996). To get an intuitive feeling of this difficulty in our problem compared to repeated second-price auctions. Let us consider the offline case where the opponents' bids are all known. Given the budget, the problem for second-price auctions can be reduced to a pure knapsack problem, where the budget is regarded as weight capacity and prices as weights. This structure enables mature techniques including duality theory to be applied to study the benchmark strategy. Pitifully in first-price auctions, since the payment depends on the buyer's own bid, the previous approach/benchmark is not directly usable. We provide a concrete example to further illustrate such difficulties.

**Example 1.1.** *Consider a case where the buyer's value $v$ follows a uniform distribution on $[0.4, 1]$ and the highest bid $m$ of her opponents' follows a uniform distribution on $[0, 0.5]$. The time horizon is $T$ and the buyer's budget $B = 0.5T$. The first-best benchmark (an anticipating[1] strategy that knows her values and her opponents' bids) can be viewed as a knapsack problem, which is*

$$\mathop{\mathbb{E}}_{\substack{\boldsymbol{v} \sim F^T \\ \boldsymbol{m} \sim G^T}} \left[ \max_{b_1, \ldots, b_T} \sum_{t=1}^{T} (v_t - b_t) \mathbf{1}_{\{b_t \geq m_t\}} \right] \quad \text{subject to} \quad \sum_{t=1}^{T} \mathbf{1}_{\{b_t \geq m_t\}} b_t \leq B \quad \forall (v_t)_{t=1}^{T}; (m_t)_{t=1}^{T},$$

*where $v_t$ is her value and $m_t$ is her opponents' highest bid at time $t$. The buyer wants to determine each $b_t$ to maximize the revenue. In hindsight, we need to pay as close to $m_t$ as possible. Using the theory of knapsack, the utility is $T \cdot \mathbb{E}[\mathbf{1}_{\{v \geq m\}}(v - m)]^+ = 0.45T$. On the contrary, the optimal non-anticipating bidding strategy in a first-price auction is to bid $\frac{v}{2}$ and the utility is $T \cdot \mathbb{E}[\mathbf{1}_{\{\frac{v}{2} \geq m\}} \frac{v}{2}] = 0.26T$. There is already an $\Omega(T)$ separation between the first-best benchmark and the ideal case with full information.*

This example shows that simple characterization of the optimum in Balseiro and Gur (2019) is not applicable to our problem. Furthermore, it remains unclear what methodology can be applied in first-price auctions with budgets. The state-of-the-art adaptive pacing strategy downgrades to truthful bidding as the budget increases, so in first-price auctions, it may result in *near-zero* reward and thus cannot have any theoretical guarantee (see (Balseiro and Gur, 2019, §2.4) for further discussions).

The present paper takes the first step to combat the challenges mentioned above with a dynamic programming approach. Correspondingly, our contribution is also two-fold:

- We provide an RL-based learning algorithm. Through the characterization of the optimal strategy, we obtain $\widetilde{O}(\sqrt{T})$-regret guarantee for the algorithm in the full-feedback case[2].

- In the censored-feedback setting, by techniques developed from survival analysis, we modify our algorithm and obtain a regret of $\widetilde{O}(T^{\frac{7}{12}})$.

## 1.1 RELATED WORK

**Repeated second-price auctions with budgets**  There is a flourishing source of literature on bidding strategies in repeated auctions with budgets. Through the lens of online learning, Balseiro and Gur (2019) identify asymptotically optimal online bidding strategies known as *pacing* (a.k.a. bid-shading in literature) in repeated second-price auctions with budgets. Inspired by the pacing strategy, Flajolet and Jaillet (2017) develop no-regret non-anticipating algorithms for learning with contextual information in repeated second-price auctions. Another line of works that uses similar techniques in the present paper includes Amin et al. (2012); Tran-Thanh et al. (2014); Gummadi et al. (2012). Gummadi et al. (2012) and Amin et al. (2012) study bidding strategies in repeated second-price auctions with budget constraints, but the former does not involve any learning and the latter does not provide any regret analysis (their estimator is biased). Tran-Thanh et al. (2014) derive regret bounds for the same scenario but the optimization objective is the number of items won instead of value or surplus. Baltaoglu et al. (2017) also use dynamic programming to tackle repeated

---

[1]An algorithm is anticipating if bid selection depends on future observations, see Flajolet and Jaillet (2017).

[2]This is especially practical in public-sector auctions (Chari and Weber, 1992) as regulations mandate all bids to be revealed.

second-price auctions with budgets. However, they assume per-round budget constraints and their dynamic programming algorithm is for allocating bids among multiple items. Again, we emphasize that no prior work has been done in repeated first-price auctions with budgets since the structure of the problem (compared to second-price variants) is fundamentally different (recall Example 1.1).

**Repeated first-price auctions without budgets**   Two notable works concerning repeated first-price auctions are Han et al. (2020b) and Han et al. (2020a). Han et al. (2020b) introduce a new problem called monotone group contextual bandits and obtain an $O(\sqrt{T}\ln^2 T)$-regret algorithm for repeated first-price auctions *without* budget constraints under stationary settings. This bound is improved to $O(T^{\frac{1}{3}+\epsilon})$ by Achddou et al. (2021) with additional assumptions on distributions. Han et al. (2020a) concentrate on an adversarial setting and develop a mini-max optimal online bidding algorithm with $O(\sqrt{T}\ln T)$ regret against all Lipschitz bidding strategies. Badanidiyuru et al. (2021) consider the problem in a contextual setting. A crucial difference is that in the present paper, budgets are involved thus the algorithms from previous works are not directly suitable for our needs.

**Bandit with knapsack**   From the bandit side, Badanidiyuru et al. (2013) develop bandit algorithms under resource constraints. They show that their algorithm can be used in dynamic procurement, dynamic posted pricing with limited supply, etc. However, since the bidder observes her value *before* bidding in our problem, results by Badanidiyuru et al. (2013) cannot be directly applied to our setting. Our setting also relates to contextual bandit problems with resource constraints (Badanidiyuru et al., 2014; Agrawal and Devanur, 2016; Agrawal et al., 2016). Nevertheless, applying this contextual bandit approach requires discretizing the action space, which needs Lipschitz continuity of distributions. Our approach does not rely on any continuity assumption. Further, the performance guarantee (typically $\widetilde{O}(T^{\frac{2}{3}})$) is worse than ours. It also does not fit into our information structure when the feedback is censored.

## 2   PRELIMINARIES

**Auction mechanism**   We consider a repeated first-price auction with budgets. Specifically, we suppose that the buyer has a limited budget $B$ to spend in a time horizon of $T \leq +\infty$ (can be *unknown* to her) rounds. At the beginning of each round $t = 1, 2, \ldots, T$, the bidder privately observes a value $v_t$ for a fresh copy of item and bids $b_t$ according to her past observations $\boldsymbol{h}_t$ and value $v_t$. Denote the strategy she employs as $\pi \colon (v_t, B_t, \boldsymbol{h}_t) \to b_t$, which maps her current budget $B_t$, value $v_t$ and past history $\boldsymbol{h}_t$ to a bid. Let $m_t$ be the maximum bid of the other bidders. Since the auction is a *first* price auction, if $b_t$ is larger than $m_t$, then the buyer wins the auction, is charged $b_t$, and obtains a utility of $v_t - b_t$; otherwise, she loses and the utility is 0. Therefore, the instantaneous utility of the buyer is

$$r_t = (v_t - b_t)\mathbf{1}_{\{b_t \geq m_t\}}.$$

The exact information structure of history the buyer observes is dictated by how the mechanism reveals $m_t$. In full generality, we assume that the feedback is censored, i.e. only the highest bid is revealed at the end of each round and the winner does not observe $m_t$ exactly. This is considered to be an informational version of *winner's curse* (Capen et al., 1971) and is of practical interest (Esponda, 2008). For the purpose of modeling, we suppose that ties are broken in favor of the buyer but this choice is arbitrary and by no means a limitation of our approach.

Next, we state the assumptions on $m_t$ and $v_t$. Without loss of generality, we assume that $b_t, m_t, v_t$ are normalized to be in $[0, 1]$. In the present paper, we consider a stochastic setting where $m_t$ and $v_t$ are drawn from some distributions $F, G$ *unknown* to the buyer, respectively, and independent from the past. We will also refer to the cumulative distribution functions of $F, G$ with the same notations. No further assumptions will be made on $F, G$. Now, the expected instantaneous utility of the buyer at time $t$ with strategy $\pi$ is

$$R^\pi(v_t, b_t) = \mathbb{E}_{m_t \sim F}[r_t] = (v_t - b_t)F(b_t).$$

To argue for the reasonability of this assumption, note that although other buyers may also involve learning behavior, it is typical that in a real advertising market, there are a large number of buyers (Kahng et al., 2004). The specific buyer only faces a different small portion of them and is completely oblivious of whom she is facing in each round. Therefore, the buyer's sole objective is to maximize

her utility (instead of fooling other buyers) and by the law of large numbers, the price $m_t$ and value $v_t$ the buyer observes are independent and identically distributed at least for a period of time[3].

**Buyer's target and regret**     The buyer aims at maximizing her long-term accumulated utility subject to budget constraints. Recall that the instantaneous utility of the buyer is $r_t = (v_t - b_t)\mathbf{1}_{\{b_t \geq m_t\}}$. The payment is $c_t = b_t \mathbf{1}_{\{b_t \geq m_t\}}$ and the budget will then decrease accordingly as the payment incurs. She can continue to bid as long as the budget has not run but must stop at

$$\tau^* = \min \left\{ T + 1, \min \left\{ t \in \mathbb{N} : \sum_{\tau=1}^{t} c_\tau = B \right\} + 1 \right\}.$$

The buyer's problem now becomes determining how much to bid in each round to maximize her accumulated utility. In line with works Gummadi et al. (2012); Golrezaei et al. (2019); Deng and Zhang (2021), the buyer adopts a discount factor $\lambda \in (0, 1)$. She takes discounts since she does not know $T$ or $\tau^*$ — Discount factors can be interpreted to be the estimate of the probability that the repeated auction will last for at least $t$ rounds (Devanur et al., 2014; Drutsa, 2018). It means that the process will terminate at each round with probability $1 - \lambda$ (Uehara et al., 2021). On the economic side, in important real-world markets like online advertising platforms, buyers are impatient for opportunities since companies of different sizes have different capabilities. Discount factors model how impatient[4] the buyer is in (Drutsa, 2017; Vanunts and Drutsa, 2019). Now the buyer's optimization problem is to determine a *non-anticipating* strategy $\pi$ for the following:

$$\max_{\pi} \quad \mathbb{E}_{\substack{\boldsymbol{v} \sim F^T \\ \boldsymbol{m} \sim G^T}} \left[ \sum_{t=1}^{T} \lambda^{t-1} r_t \right] \quad \text{subject to} \quad \sum_{t=1}^{T} \mathbf{1}_{\{b_t \geq m_t\}} b_t \leq B \quad \forall (v_t)_{t=1}^{T}; (m_t)_{t=1}^{T},$$

where $b_t = \pi(v_t, B_t, \boldsymbol{h}_t)$. Here, $\boldsymbol{v} := (v_1, \ldots, v_T)$ denotes the sequence of private values the buyer observes, and $\boldsymbol{m} := (m_1, \ldots, m_T)$ is the sequence of the highest bids of the other bidders. $V^{\pi}(B, t)$ denotes the expected accumulated utility using strategy $\pi$ with budget $B$ and starting from time $t$. Let $\pi^*$ denote the optimal bidding strategy with the knowledge of the underlying distributions $F$ and $G$. The corresponding expected accumulated utility is $V^{\pi^*}(B, t)$. (We sometimes use $V(\cdot, \cdot)$ to represent $V^{\pi^*}(\cdot, \cdot)$ for convenience in the rest of the paper.)

We now come to define the regret. First, write the per-episode revenue suboptimality for each round $t$ as

$$\text{SubOpt}_t(\pi_t) = V^{\pi^*}(B_t, t) - V^{\pi_t}(B_t, t),$$

where $\pi_t$ is the strategy used in round $t$. Our evaluation metric is then the sum of suboptimality for $t = 1, \ldots, T$, namely

$$\text{Regret}(T) = \mathbb{E} \left[ \sum_{t=1}^{T} \text{SubOpt}_t(\pi_t) \right], \tag{1}$$

where the expectation is taken over the trajectories of the achievement of $\boldsymbol{v}$ and the realization of others' bids inexplicitly.

The definition of regret comes from traditional reinforcement learning (RL) literature of infinite-horizon discounted model (Kaelbling et al., 1996). The definition is also inspired by the recent advances in Yang et al. (2021); He et al. (2021); Liu and Su (2020); Zhou et al. (2021). Zhou et al. (2021) call it cumulative error. It reflects the suboptimality for $\pi_t$ to learn the optimal valuation of attending the auction.

In the most common scenario, such as Balseiro and Gur (2019), the budget constraint is linear to time horizon $T$, i.e. $\frac{B}{T} \sim \Theta(1)$. Therefore, a bidder has an expectation that she will win for $O(T)$ rounds.

---

[3]This assumption has support from experimental evidence (Pin and Key, 2011). It can also be theoretically justified using mean field asymptotics. Please also see Han et al. (2020b) for another justification.

[4]As an additional explanation, in budget-constrained first-price auctions, the bidder always bids below or equal to her value. So she is very sensitive to the market price. However, by not winning the auction at a certain price, the bidder creates a future opportunity to win an equivalent auction at a lower price. The use of a bid discount factor adds flexibility to tune this behavior. As the bidder has a preference for present utility over future utility, the discount factor moderates the extent of underbidding that she finds to be optimal, which makes the model more general.

With a sub-optimal policy, it is easy to suffer $O(T)$ regret which is intolerable for bidders. It leads to the challenge to achieve a sublinear regret in first-price auctions and we design algorithms to answer the question.

## 3 BIDDING ALGORITHM AND ANALYSIS

In this section, we present our bidding algorithm and the high-level ideas in the analysis of regret. We first consider the case where the feedback is not censored, i.e. the buyer is aware of $m_t$ no matter whether she wins or not. Then we extend our algorithm to the case where the feedback is censored with techniques developed from survival analysis.

### 3.1 FULL FEEDBACK

When $F$ and $G$ are known, the buyer's problem can be viewed as offline. The technical challenge lies in the observation that even when the distributions are known, the buyer's problem cannot be directly analyzed as a knapsack problem. To tackle this challenge, we use a dynamic programming approach to solve the problem. In particular, the optimal strategy $\pi^*$ satisfies the following Bellman equation:

$$b^*(B_\tau, v) \in \arg\max_b [(v - b) + \lambda V(B_\tau - b, \tau + 1)] F(b) + \lambda V(B_\tau, \tau + 1)(1 - F(b)),$$

$$V(B_\tau, \tau) = \mathbb{E}_v [(v - b^*) + \lambda V(B_\tau - b^*, \tau + 1)] F(b^*) + \lambda V(B_\tau, \tau + 1)(1 - F(b^*)),$$

for all $\tau \in \mathbb{N}$ and $0 \leq B_\tau \leq B$. Note that for any $B_\tau < 0$, $V(B_\tau, \tau) = -\infty$. By choosing appropriate initialization conditions, we can solve the equation recursively and obtain the optimal bidding strategy together with the function $V(\cdot, \cdot)$. The above recursion also characterizes the optimal solution, which will be used in the analysis later.

When the buyer does not have the information of $F$ and $G$, she can learn the distributions from past observations. Therefore, it is natural to maintain estimations $\hat{F}$ and $\hat{G}$ of the true distributions. Our algorithm for the full-feedback case is now depicted in Algorithm 1. To ease technical loads, we first assume the knowledge of $G$ and only estimate $F$ in Algorithm 1. Later, we will add the estimation of $G$ and its analysis is presented in Theorem 3.2.

---

**Algorithm 1** Algorithm for the full-feedback case

1: **Input**: Initial budget $B$ and constant $C_1$          ▷ $C_1$ is an arbitrary positive constant
2: Initialize the estimation $\hat{F}$ of $F$ to a uniform distribution over $[0, 1]$ and $B_1 \leftarrow B$
3: **for** $t = 1, 2, \ldots$ **do**
4:      Observe the value $v_t$ in round $t$
5:      Let $t_0$ be the smallest integer that satisfies $\lambda^{t_0 - t} \frac{1}{1 - \lambda} < \frac{C_1}{\sqrt{t}}$
6:      Set $V_{\hat{F}}(B_{t_0}, t_0) = 0$ for any $B_{t_0}$          ▷ $V_{\hat{F}}$ is algorithm's estimation of $V$
7:      **for** $\tau = t_0, t_0 - 1, \ldots, t$ **do**
8:          $Q_{v, \hat{F}}(B_\tau, \tau, b) \leftarrow [(v - b) + \lambda V_{\hat{F}}(B_\tau - b, \tau + 1)] \hat{F}(b) + \lambda V_{\hat{F}}(B_\tau, \tau + 1)(1 - \hat{F}(b))$
9:          Solve the optimization problem $\hat{b}_\tau^* \leftarrow \arg\max_b Q_{v, \hat{F}}(B_\tau, \tau, b)$
10:         $V_{\hat{F}}(B_\tau, \tau) \leftarrow \mathbb{E}_{v \sim G}[Q_{v, \hat{F}}(B_\tau, \tau, \hat{b}_\tau^*)]$
11:      **end for**
12:      Place a bid $\hat{b}_t \leftarrow \arg\max_b Q_{v, \hat{F}}(B_t, t, b)$
13:      Observe $m_t, c_t$ and $r_t$ from this round of auction and update $\hat{F}(x) = \frac{1}{t} \sum_{i=1}^{t} \mathbf{1}_{\{m_i \leq x\}}$.
14:      $B_{t+1} \leftarrow B_t - c_t$. If $B_{t+1} \leq 0$ then halt.
15: **end for**

---

Similar to prior work (Amin et al., 2012), Algorithm 1 performs exploration and exploitation simultaneously. The buyer initializes the estimation of $F$ to a uniform distribution (Line 2). At round $t$, the buyer observes her valuation. Then, she uses her estimation of $F$ to solve the dynamic programming problem recursively[5] to obtain an estimation of the optimal bid (Line 7~Line 11). To

---

[5]For the non-trivial case $B \leq T$, this can be solved in $O\left(\frac{T^{4.5}}{(1 - \lambda)^6}\right)$ time with only $O(T^{-\frac{1}{2}})$ loss (see, e.g. Chow and Tsitsiklis, 1989).

provide a base case for recursion, note that for sufficiently large $t_0 \gg t$, $V_{\hat{F}}(\cdot, t_0)$'s impact to $V_{\hat{F}}(\cdot, t)$ is negligible due to the discount $\lambda^{t_0 - t}$. Therefore, the buyer can approximate $V_{\hat{F}}(\cdot, t_0)$ with zero for $t_0$ (Line 5). Finally, the auction proceeds with $m_t, r_t, c_t$ revealed and the buyer updates her information accordingly (Line 13~Line 14).

**Analysis of regret**  To analyze the algorithm, we first assume that Algorithm 1 knows the distribution $G$ exactly and establishes the regret. Then we add the contribution of the estimation of $F$.

**Theorem 3.1.** *Under the circumstance that $F$ is unknown, the worst-case regret of Algorithm 1 is $\widetilde{O}(\sqrt{T})$, where the regret is computed according to Equation* (1). *Explicitly, if we take $C_1 = 1$,*

$$Regret(T) \leq \left( 4\sqrt{\ln(\sqrt{2}T)}\frac{1+\lambda}{(1-\lambda)^3} + \frac{4}{1-\lambda} \right) \sqrt{T} + 1.$$

To show an example of the application of the result, let us take the budget $B$ to scale linearly with $T$ as in Balseiro and Gur (2019); Flajolet and Jaillet (2017). Specifically, assume that $T < +\infty$ and there exists a constant $\beta$ such that the budget $B = \beta T$, then we establish that the regret is $\widetilde{O}(\sqrt{T})$ in this special case. Indeed, under this condition, we can simply set $t_0 = T + 1$ and $V_{\hat{F}}(B_{T+1}, T + 1) = 0$ for any $B_{T+1}$ in Algorithm 1. Therefore, $C_1 = 0$ and the worst-case regret is bounded by $\left( 4\sqrt{\ln(\sqrt{2}T)}\frac{1+\lambda}{(1-\lambda)^3} \right) \sqrt{T} + 1$.

Next, we deal with the case where $G$ is also initially unknown. Based on Algorithm 1, we additionally maintain an estimation $\hat{G}$ of $G$ based on past observations of valuations. $\hat{G}$ is initialized to be a uniform distribution and will be used to solve the dynamic programming problem (see Line 7 of Algorithm 2). Using similar techniques as before (with more work), we obtain the following theorem.

**Theorem 3.2.** *Under the circumstance that $F, G$ are both unknown, it holds that the worst-case regret of Algorithm 1 using empirical distribution functions to estimate $F$ and $G$ is $\widetilde{O}(\sqrt{T})$. Explicitly, if we take $C_1 = 1$,*

$$Regret(T) \leq \left( \sqrt{\ln(2T)}\frac{6(1+\lambda)}{(1-\lambda)^3} + \frac{4}{1-\lambda} \right) \sqrt{T} + 1.$$

### 3.2 CENSORED FEEDBACK

In this subsection, we deal with the case that the buyer can only see the winner's bid after each round. In other words, the feedback is left-censored. Concretely, the buyer's observation is

$$o_t = \max\{b_t, m_t\}.$$

When she wins, the exact value of $m_t$ is not revealed. The buyer only knows that $m_t$ is smaller than her bid in the current round. To estimate the distribution of $m_t$, there is a classical statistics (KM estimator) developed by Kaplan and Meier (1958) for the estimation of $F$ in this scenario. However, the KM estimator assumes the sequence $(m_t)_{t=1}^T$ is deterministic, which does not fit our needs. Although Suzukawa (2004)'s extension allows random censorship, it requires independence between $b_t$ and $m_t$, which is not realistic since we use the estimated distribution to place bids.

To tackle this problem, we first introduce an estimator proposed by Zeng (2004) denoted by $\hat{F}_n$ to take place of the previous empirical distribution used in Algorithm 1.

**Estimation procedure**  We now present the procedure for estimating $F$ under censored feedback. It suffices to estimate the distribution function of $1 - m_t$ which is right-censored by $1 - b_t$. Let $y_t = \min\{1 - m_t, 1 - b_t\}$, $r_t = \mathbf{1}_{\{m_t \geq b_t\}}$. The observations can now be described as $(y_t, r_t, \boldsymbol{h}_t)_{t=1}^T$.

Roughly speaking, to decouple the dependency between $m_t, b_t$, we use the fact that $b_t$ and $m_t$ are independent conditioning on $\boldsymbol{h}_t$. Intuitively, the history $\boldsymbol{h}_t$ provides information for getting enough effective samples for $m_t$. Next, we establish models to estimate the hazard rate functions[6] of $1 - m_t, 1 - b_t$ using $\boldsymbol{h}_t$. With the hazard rate functions, we use the maximum likelihood method with a kernel to compute the final estimation $\hat{F}_t$ and obtain Equation (3).

---

[6]The hazard rate function of a random variable $X$ with p.d.f. $f$ and c.d.f. $F$ is $H_X(x) = \frac{f(x)}{1-F(x)}$.

Details follow. We initialize with a sequence (bandwidth) $(a_t)_{t=1}^T$ such that $\frac{\log^2 a_t}{t a_t^2} \to 0, t a_t^2 \to \infty, t a_t^4 \to 0$ as $t \to +\infty$ and a symmetric kernel function $K(\cdot, \cdot) \in C^2(\mathbb{R}^2)$ with bounded gradient. Now, at each time $t$, we compute two vectors $\boldsymbol{\beta}_t, \boldsymbol{\gamma}_t$ which maximize each of the following likelihood functions (these can be regarded as loss functions of the estimation that we aim to optimize)

$$f(\boldsymbol{\beta}) = \sum_{\tau=1}^t \frac{r_\tau}{t} \left( \boldsymbol{\beta}^\top \boldsymbol{h}_\tau - \log \sum_{y_i \geq y_\tau} e^{\boldsymbol{\beta}^\top \boldsymbol{h}_\tau} \right), g(\boldsymbol{\gamma}) = \sum_{\tau=1}^t \frac{1 - r_\tau}{t} \left( \boldsymbol{\gamma}^\top \boldsymbol{h}_\tau - \log \sum_{y_i \geq y_\tau} e^{\boldsymbol{\gamma}^\top \boldsymbol{h}_\tau} \right). \tag{2}$$

We arbitrarily pad $\boldsymbol{h}_1, \ldots, \boldsymbol{h}_t$ with zeros to make their length the same (we will show that this is without loss of generality in a moment). Compute $\boldsymbol{Z}_t = (\boldsymbol{\beta}_t^\top \boldsymbol{h}_t, \boldsymbol{\gamma}_t^\top \boldsymbol{h}_t)^\top$. The survival function of $1 - m_t$, or equivalently the cumulative distribution function of $m_t$, is estimated based on Zeng (2004)'s estimator

$$\hat{F}_t(x) = \frac{1}{t} \sum_{i=1}^t \prod_{j=1}^t \left( 1 - \frac{K((\boldsymbol{Z}_i - \boldsymbol{Z}_j)/a_n) \mathbf{1}_{\{y_j \leq x\}} r_j}{\sum_{m=1}^n K((\boldsymbol{Z}_i - \boldsymbol{Z}_m)/a_n) \mathbf{1}_{\{y_j \leq y_m\}}} \right). \tag{3}$$

Now, we are ready to apply the estimator to design the algorithm for the censored-feedback case. Note that the new estimator's convergence rate is slower than that for the full-feedback case. Therefore, compared to Algorithm 1, Algorithm 2 is now a multi-phase algorithm. The algorithm only updates the estimation of $\hat{F}$ at the end of each phase (see Figure 1 for an illustration). The other elements of each phase in Algorithm 2 are similar to Algorithm 1.

---

**Algorithm 2** Algorithm for the censored-feedback case

---

1: **Input**: Initial budget $B$ and constant $C_1$        $\triangleright C_1$ is an arbitrary positive constant
2: Initialize the estimation $\hat{F}$ of $F$ and the estimation $\hat{G}$ of $G$ to uniform distributions over $[0, 1]$
3: $B_1 \leftarrow B$
4: **for** Phase $i = 1, 2, \ldots$ **do**      $\triangleright$ Phase $i$ $(i > 1)$ lasts for $2^i$ rounds. Phase 1 lasts for 2 rounds
5:     **for** each $t$ in the time interval of round $i$ **do**
6:        Observe the value $v_t$ in round $t$
7:        Update $\hat{G}(x) = \frac{1}{t} \sum_{i=1}^t \mathbf{1}_{\{v_i \leq x\}}$.
8:        Let $t_0$ be the smallest integer that satisfies $\lambda^{t_0 - t} \frac{1}{1-\lambda} < \frac{C_1}{\sqrt{t}}$
9:        Set $V_{\hat{F}, \hat{G}}(B_{t_0}, t_0) = 0$ for any $B_{t_0}$      $\triangleright V_{\hat{F}, \hat{G}}$ is algorithm's estimation of $V$
10:        **for** $\tau = t_0, t_0 - 1, \ldots, t$ **do** $\triangleright$ This loop can be moved to the end of each phase to reduce the invocation time from $T$ to $\ln T$
11:           $Q_{\hat{F}, \hat{G}}(B_\tau, \tau, b) \leftarrow [(v - b) + \lambda V_{\hat{F}, \hat{G}}(B_\tau - b, \tau + 1)]\hat{F}(b) + \lambda V_{\hat{F}, \hat{G}}(B_\tau, \tau + 1)(1 - \hat{F}(b))$
12:           Solve the optimization problem $\hat{b}_\tau^* \leftarrow \arg\max_b Q_{\hat{F}, \hat{G}}(B_\tau, \tau, b)$
13:           $V_{\hat{F}, \hat{G}}(B_\tau, \tau) \leftarrow \mathbb{E}_{v \sim G}[Q_{\hat{F}, \hat{G}}(B_\tau, \tau, \hat{b}_\tau^*)]$
14:        **end for**
15:        Place a bid $\hat{b}_t \leftarrow \arg\max_b Q_{\hat{F}, \hat{G}}(B_t, t, b)$
16:        Observe $o_t, c_t$ and $r_t$ from this round of auction
17:        $B_{t+1} \leftarrow B_t - c_t$. If $B_{t+1} \leq 0$ then halt.
18:     **end for**
19:     Update $\hat{F}$ using the estimator specified in Equation (3) with data observed before this phase
20: **end for**

---

**Analysis of regret** To analyze the performance of Algorithm 2, we will prove a series of lemmas on the estimation error of Equation (3). We concentrate on the performance of the new estimator since this is the major difference between Algorithm 1 and Algorithm 2. In particular, our proof relies on the following convergence result.

**Lemma 3.3** (Zeng). *Let $\hat{F}_n$ be the estimation of $F$ after using $n$ observations. We have*

$$\sqrt{n}(\hat{F}_n(1 - x) - F(1 - x)) \Longrightarrow \mathcal{B}(x) \quad in \quad \ell^\infty([0, 1]),$$

*where $\mathcal{B}(x)$ is a Gaussian process.*

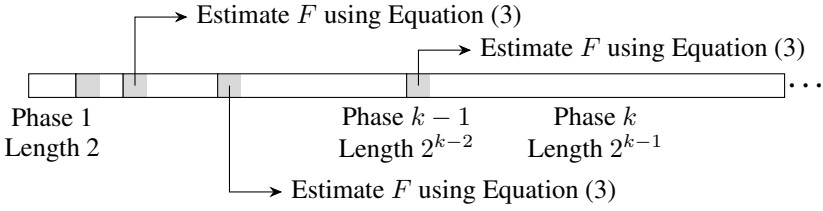

Figure 1: Schematic representation of the phases in Algorithm 2. Algorithm 2 updates its estimates of $F$ at the end of each phase.

Before we proceed to apply the lemma, we verify a series of prerequisites mentioned in Zeng (2004) to make sure it holds. First, we make sure that conditioning on $\boldsymbol{h}_t$, the random variables $1 - m_t$ and $1 - b_t$ are independent. Indeed, $b_t$ is completely decided by $\boldsymbol{h}_t$ and $m_t$ is independent of $\boldsymbol{h}_t$. Second, we note that the maximizer shown in Equation (2) is essentially doing Cox's proportional hazards regression analysis. To establish consistency of the estimator, we show that at least one of $\widetilde{m}_t := 1 - m_t$ and $1 - b_t$ follows Cox's proportional model. That is to say, there exists $\boldsymbol{\beta}$ and a function $f(y)$ such that the hazard rate function of $\widetilde{m}_t$ or $1 - b_t$ conditioning on $\boldsymbol{h}_t$ exactly follows

$$H(y \mid \boldsymbol{h}_t) = f(y)e^{\boldsymbol{\beta}^\top \boldsymbol{h}_t}. \tag{4}$$

Equation (4) holds for $\widetilde{m}_t$. In fact, taking $\boldsymbol{\beta} = \boldsymbol{0}$ and $f(y) = \frac{F'(1-y)}{1-F(1-y)}$ suffices. Since we take $\boldsymbol{\beta} = \boldsymbol{0}$, consistency establishes regardless of the way we pad $\boldsymbol{h}_t$.

Next, consider some phase at time $2^n \leq t \leq 2^{n+1} - 1$. The estimation $\hat{F}_n$ is computed using the first $2^n$ observed data points. Applying similar techniques for the rate of convergence of empirical process (Norvaiša and Paulauskas, 1991), we obtain the following lemma on the performance of $\hat{F}$ in Algorithm 2.

**Lemma 3.4.** *Under the update process in Algorithm 2, for any $2^n \leq t \leq 2^{n+1} - 1$, we have the following bounds for the estimation $\hat{F}_n$:*

$$|\Pr(\sup_b |\sqrt{2^n}(\hat{F}_n(1-b) - F(1-b))| \geq r) - \Pr(\sup_b |\mathcal{B}(1-b)| \geq r)| \leq M(1+r)^{-3}\ln^2(t) \cdot t^{-\frac{1}{6}},$$

*where $M$ is a constant depending only on $F$ and Algorithm 2.*

Finally, we now bound the difference between $\hat{F}_n$ and $F$ with the help of Lemma 3.4.

**Lemma 3.5.** *Recall that we use the first $2^n$ data points to estimate $\hat{F}$. Under the update procedure of Algorithm 2, for any $2^n \leq t \leq 2^{n+1} - 1$, with probability at least $1 - T^{-\frac{5}{12}}/(2\ln T)$*

$$\sup_x |\hat{F}(x) - F(x)| \leq \sqrt{2}C_5(4M\ln^3 T)^{\frac{1}{3}}t^{-\frac{5}{9}}T^{\frac{5}{36}},$$

*where $C_5$ is an absolute constant.*

With Lemma 3.5 in hand, we now have

**Theorem 3.6.** *Under the circumstance that $F, G$ are both unknown and the feedback is censored, the worst-case regret of Algorithm 2 is $\widetilde{O}(T^{\frac{7}{12}})$. Explicitly, if we take $C_1 = 1$,*

$$Regret(T) \leq \left(\frac{9\sqrt{2}(1+\lambda)}{2(1-\lambda)^3}C_5(4M\ln^3 T)^{\frac{1}{3}} + 1\right)T^{\frac{7}{12}} + \left(\sqrt{\frac{1}{2}\ln(4T^{\frac{17}{12}})}\frac{2(1+\lambda)}{(1-\lambda)^3} + \frac{4}{1-\lambda}\right)\sqrt{T}.$$

**Remark 3.7.** *The setting in Han et al. (2020b) is a special case of ours, where there are no budget constraints and $\lambda = 0$ (thus removing the $\frac{1}{1-\lambda}$ factor in our results). The buyer's aim is to maximize $(v - b)F(b)$ each round. This is equivalent to $V_{\hat{F}} = 0$ in our setting with no need to estimate $G$, yielding regret $\widetilde{O}(\sqrt{T})$ in the full-feedback case and regret $\widetilde{O}(T^{\frac{7}{12}})$ in the censored-feedback case.*

**Remark 3.8.** *The regret-bound looks unusual at a first glance. The reason is that the convergence rate of the estimator is lower than that in the commonly used "Hoeffding-type" or "Bernstein-type" inequalities. However, due to the information structure, they are not suitable to be used in our environment to our best knowledge.*

## 4 LOWER BOUND

Here, we discuss the lower bound for the regret under such settings. This will shed light on the optimality gap of the proposed policies.

**Full Feedback**   We have proposed a general solution framework that works for any $\beta$ where the budget constraint is $B = \beta T$ and any discount rate $\lambda \in [0, 1)$. Note that the general lower bound is no less than the lower bound for a specific case. Consider the case when $\beta = 1$ and $\lambda = 0$. Our problem reduces to the case where the buyer essentially does not face budget constraints and is extremely myopic. Under this circumstance, the problem is a multi-armed bandit problem. Auer et al. (2002) shows that it suffers from a $\Theta(\sqrt{T})$-regret lower bound. This means that our algorithm is *optimal* up to logarithmic terms.

**Censored Feedback**   The $\widetilde{O}(\sqrt{T})$ lower bound also applies here. However, the upper bound and the lower bound have not matched yet. We leave this as an intriguing open problem as there is a lack of relevant literature to show regret lower bounds under the censored-feedback case. However, we want to provide some evidence that our upper bound is sufficiently good. For example, parallel to our work, Gaitonde et al. (2022) extend the pacing techniques to a class of auction forms including first-price auctions. They obtain an $\widetilde{O}(T^{\frac{3}{4}})$-regret bounds against *the best linear policy* under the *value-maximization* objective. Under a censored-feedback information structure with contextual valuations, Cesa-Bianchi et al. (2017) show an $\widetilde{O}(T^{\frac{d-1/3}{d+2/3}})$-regret upper bound without budget constraints where $d$ is the dimension of the context. And a similar information and payment structure in Bayesian persuasion yield an $\widetilde{O}(T^{\frac{4}{5}})$ regret bound (Castiglioni et al., 2020).

## 5 DISCUSSION AND CONCLUSION

In this paper, we develop a learning algorithm to adaptively bid in repeated first-price auctions with budgets. On the theoretical side, our algorithm, together with its analysis of $\widetilde{O}(\sqrt{T})$-regret in the full-feedback case and $\widetilde{O}(T^{\frac{7}{12}})$-regret in the censored-feedback case, takes the first step in understanding the problem. On the practical side, our algorithm is simple and readily applicable to the digital world that has shifted to first-price auctions[7].

Questions raise themselves for future explorations. We observe here that in the limiting case $\lambda \to 1$, the optimal bidding strategy in Algorithm 2 is similar to a *pacing* strategy, which relates to the open question[8] raised in Balseiro and Gur (2019). In the limiting case of $\lambda \to 1$, the optimal bid of Algorithm 2 is of the form $\frac{v_t}{1+x_t}$, where $x_t$ is a pacing multiplier that depends only on $B_t$ and $F$ and can be computed without solving the dynamic programming problem. This observation can be viewed as a corollary of (Theorem 3.1 Gummadi et al., 2012). This connection between Algorithm 2 and pacing suggests further investigations. Other immediate open questions include closing the gap between upper and lower bounds for the censored feedback case.

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
