# OpenReview forum: "No-regret Learning in Repeated First-Price Auctions with Budget Constraints"
_ICLR.cc/2023/Conference — Submitted to ICLR 2023_

### Official Review · Reviewer_fj85 · 2022-10-20

**Confidence:** 2
**Correctness:** 3
**Technical Novelty And Significance:** 1
**Empirical Novelty And Significance:** 1
**Recommendation:** 5

**Clarity, Quality, Novelty And Reproducibility:**

I think it's neither very clear (see comments below), nor very novel (mostly combining known techniques)

**Strength And Weaknesses:**

•	I’m a bit confused by the way the discount factor is presented:

o	In the context of a global budget across time periods, I’m not sure what the discount factor models.

o	I cannot find where in the definition of SubOpt the discount factor appears

o	On the experiments, I would like to see what happens with a more aggressive discount factor

•	I believe that the term “winner’s curse” is used incorrectly. It just means that winners in auctions tend to be the ones who overestimate the true value of the item for sale, so they’re likely to actually make a bad deal.

•	Are F and G independent from each other? (And if so, what does that model?

•	You should clearly define non-anticipating strategies

•	I’m not sure that the solution to the DP is exactly the “optimal” bidding strategy as you claim: AFAICT it doesn’t account for the noise – e.g. it only satisfies the budget in expectation.

•	Typo: “worse-case”

•	$\frac{1}{2}\ln(T^2)$  --  Wouldn’t it be easier to cancel the 2 and the 1/2?

•	Does the DP really take you T^4.5 time to solve? What did you do in the experiments?

•	I couldn’t follow (at a high level) what are f and g.

•	What does the \implies symbol in Lemma 3.3 mean?

•	In Lemma 3.4, you want to bound the sup of the learnt and actual distribution. Maybe that’s too strong? You could get a better result by relaxing that?

•	“regret bound is the best possible…” – can you formalize that claim?

•	Did you use T=2,000,000 or T=200,000?

•	The ~5% regret in experiments seems pretty high.
o	\sqrt{T} is roughly 1,000, so that’s 1+ order of magnitudes off
o	In the worst case theorem there are some hidden constants, but this is on pretty reasonable distributions and very large T


**Summary Of The Paper:**

The submission proposes an online learning approach to bidding in repeated stochastic first price auctions with a global budget constraint. The main idea is to approximately learn the distribution of values and other bidders’ top bid, and then solve a DP for the optimal “non-anticipating” strategy assuming values and bids will actually be drawn from learnt distributions.

**Summary Of The Review:**

The problem seems of fairly wide interest, but the results given in the submission are not particularly exciting and the techniques aren't novel AFAICT.

---

### Official Review · Reviewer_6CuB · 2022-10-24

**Confidence:** 2
**Correctness:** 2
**Technical Novelty And Significance:** 3
**Empirical Novelty And Significance:** 3
**Recommendation:** 6

**Clarity, Quality, Novelty And Reproducibility:**

This work gets some new results for first price auction in repeated settings. The motivation and results are clear. But the writing for theory part is not very good, I can not get either the clear proof  or good intuition examination.

**Strength And Weaknesses:**

Strength: This paper initiates the study to learn online bidding strategies in repeated first-price auctions with budgets and shows good sublinear regret.

Weakness: No proof and less explanations for the theorem and equation used (such as the meaning of the Bellman equation used in this paper), so it is quite hard to check the correctness.
I have an question about the theorem. In theorem 3.1, from the inequation, the regret is \tilde O (T), but not the \tilde O (T^{1/2}) shown in the context. Similarly for Theorem 3.2 and 3.6
Typo: Theorem 3.2 should be Section 3.2 on Page 5



**Summary Of The Paper:**

This paper initiates the study for a buyer with budgets to learn online bidding strategies in repeated first-price auctions. They propose an RL-based bidding algorithm against the optimal non-anticipating strategy under stationary competition. The algorithm obtains T^{1/2}-regret if the bids are all revealed at the end of each round. With the restriction that the buyer only sees the winning bid after each round, the algorithm obtains T^{7/12}-regret by techniques developed from survival analysis.

**Summary Of The Review:**

This paper studies for buyer with budgets to learn online bidding strategies in repeated first-price auctions. The contribution is quite good, but I think the writing should be improvement. I can’t check the correctness. I have some problems about the Theorem results, the bound in the theorem is not consistent with the result in the context.

I vote for a weakly reject as it hasn’t convince me (mainly because of the writting), although I think the results are quite good.

---

### Official Review · Reviewer_pUFT · 2022-11-02

**Confidence:** 3
**Correctness:** 3
**Technical Novelty And Significance:** 2
**Empirical Novelty And Significance:** Not applicable
**Recommendation:** 6

**Clarity, Quality, Novelty And Reproducibility:**

The following comments are rather minor. I would appreciate more if they can be addressed and polished in the paper.
1. The notation of $\tilde{O}$ is used throughout the paper, without a clear definition specifying what it hides or if it has a same meaning in different locations. The section of related work has a notation of $O$ carrying polylog terms. This potentially delivers a feeling that $\tilde{O}$ hides more than polylog terms.
2. For Lemma 3.4, the statement of '$M$ is a constant depending only on $F$ and Algorithm 2' is vague. Especially, it is unclear to what extend '$M$ depends on Algorithm 2. Can you elaborate further?
3. For Lemma 3.5 and Theorem 3.6, where the undefined notation $C_5$ appears, I understand there should be a series of $C_i$ in the appendix, but a good practice should be always keeping the main text in a self-contained shape.

**Strength And Weaknesses:**

Strength:
1. The paper is well organized and easy to read overall. The difference between the two scenarios is clearly addressed.
2. Good discussion on why the prior works do not work for this new setting.

Weaknesses:
1. It it somewhat disappointing that there is not any lower bound result. Given the model is claimed to be new, I am fine with any conjecture discussion on the lower bound. Without a lower bound, it is hard to see if the upper bound is nontrivial.
2. Some arguments in the paper are lack of proper citations for support. i). Page 3: 'it is typical that in a real advertising market....'. ii). Page 8: 'Note that by domain knowledge...'

**Summary Of The Paper:**

The paper considers the problem of repeated first-priced auctions with budget constraints, a new model that did not capture attention before. Two scenarios of full feedback and censored feedback are analyzed. For full feedback, the proposed algorithm has a regret of $\tilde{O}(\sqrt{T})$. For censored feedback,  the proposed algorithm has a regret of $\tilde{O}(T^{\frac{7}{12}})$

**Summary Of The Review:**

The score is mainly based on the first item of the weaknesses. I will consider raising the score if the concerns above can be addressed.

---

### Official Review · Reviewer_NGYd · 2022-11-02

**Confidence:** 2
**Correctness:** 3
**Technical Novelty And Significance:** 3
**Empirical Novelty And Significance:** 2
**Recommendation:** 6

**Clarity, Quality, Novelty And Reproducibility:**

The paper is clear and well-structured. The paper presents a novel analysis for an algorithm working on a well-established setting (first-price auction).

**Strength And Weaknesses:**

The paper is mainly theoretical and presents only a few empirical experiments. I think that the topics covered by this work are interesting from an applied point of view.

The experimental part would have benefited from a real-world example. In particular:
"Note that by domain knowledge, most valuations and bids in online auctions follow exponential distributions, so we believe that our simulations can correctly reflect real scenarios". This statement is not supported by any evidence. I would instead say that the empirical evaluation is preliminary and on synthetically generated data. I think that without evidence you cannot claim that your experiments reflect the real phenomenon of bidding.

Why not use the dataset with full feedback as a real-world benchmark?

Minor:
worse-case -> worst-case
Choosing a name for the algorithm would help to refer it to in the experimental section
Regarding the \lambda parameter, did you run a sensitivity analysis on this parameter to understand its impact on the algorithm?

**Summary Of The Paper:**

The paper studies the first price auction in a repeated setting and with budget constraints. The authors propose some algorithms showing sublinear regret under different levels of knowledge of the learner in the setting. The authors also provide some synthetically generated experiments to check the performances of the algorithm.

**Summary Of The Review:**

The paper presents a new algorithm for first-price auctions and provides theoretical guarantees on the regret. The empirical analysis is quite preliminary.

---

### Official Review · Reviewer_s5bF · 2022-11-03

**Confidence:** 4
**Correctness:** 1
**Technical Novelty And Significance:** 2
**Empirical Novelty And Significance:** Not applicable
**Recommendation:** 3

**Clarity, Quality, Novelty And Reproducibility:**

The paper is clearly written and the result seems novel. I said in my earlier point, however, the model studied does not seem adapted.


**Strength And Weaknesses:**

**Strengths:**
- The problem studied (learning first-price auction with budget) is of importance, is actual and is challenging.
- The results seems non-trivial technical and provide non-trivial results

**Weakness:** unless I missed something, I do not think that the regret definition used by the authors is adapted to the problem. It seems to me that designing a low regret algorithm for this problem is not a hard task. This leans me to reject the paper.

More precisely: the authors introduce an artificial discounted first auction problem with budget constraint with a given discount factor \lambda<1 that does not depend on the budget B. My problem is that I have the impression that when B is large $(B\gg 1/(1-\lambda)$, the optimal policy essentially does not depend on B because the budget will never expire before $\lambda^t$ becomes too small. To me, this means that a learning algorithm that completely ignore the budget will have a sublinear regret.



**Summary Of The Paper:**

This paper presents a learning algorithm for first price auction with budget constraint. The authors consider an artificial discounted scenario and measure the regret of a learning algorithm as the sum of the suboptimality gap. Their propose two algorithms: the first always observe the highest bid of the others while the second observes the highest bid of others only when it does not win. They show that their algorithm have a sublinear regret.


**Summary Of The Review:**

The paper is interesting but it does not prove what is claimed, because the regret definition is not adapted.

---

### Official Review · Reviewer_NJbs · 2022-11-04

**Confidence:** 3
**Correctness:** 4
**Technical Novelty And Significance:** 3
**Empirical Novelty And Significance:** 3
**Recommendation:** 8

**Clarity, Quality, Novelty And Reproducibility:**

Clarity, novelty and reproducibility -- The paper ranks does well in all of these categories.

**Strength And Weaknesses:**

Strength -- An important problem, very well written paper with the final algorithm built up using simpler ones working under simplifying assumptions.

Weakness --

1. The buyer interacts with a stochastic environment as opposed to a strategic one. The paper justifies this assumption in the case of large internet market places where due to law of large number effects, the strategic actions of one buyer does not affect the rest.

2. No insights into any lower bounds/reductions. The paper however sheds some light by showing that comparing with the hindsight anticipatory optimal strategy is not productive as even simple cases yields linear regret. Nevertheless, a more principled discussion on lower bounds or reductions is a nice to have.

That said, the strengths in the paper outweighs the weaknesses.


**Summary Of The Paper:**

The paper studies the interesting problem of repeated bidding in a stochastic first price auction setting with budget constraints. The paper builds up to the algorithm by first presenting and analyzing the simpler case of (i) only the player's reward distribution is unknown, and (ii) feedback is not censored by the winner's curse. Under these simplifying assumptions, the paper presents a simple algorithm -- estimate the unknown F and bid by solving the Bellman recursion for the optimal bid by using the estimated distribution. The paper then builds up to an elegant phase based solution for the case when the simplifying assumptions are not there. The winner's curse introduces bias in the estimation that the paper proposes to solve the one-sided bias through a kernelized estimator to estimate the hazard function of the unknown distribution. Since the rate of convergence of this estimator is slower, the algorithm is layered and only updates the unknown estimates in phases of exponentially increasing lengths. This phase based algorithm also gives the statistical independence between the estimator's error and the regret, that can be used for analysis.

**Summary Of The Review:**

A well written paper with clear insights and novel contributions on an important problem.

---

### Decision · Program_Chairs · 2023-01-20

**Decision:**

Reject

**Justification For Why Not Higher Score:**

The paper lacks too many components to be accepted. Lower bound, clarity, definition, rigor, comparison with literature

**Justification For Why Not Lower Score:**

N/A

**Metareview: Summary, Strengths And Weaknesses:**

This paper looks at a variant of repeated first-price auctions, with the extra-assumption that there is a budget to be respected.

The regret definition is quite weird and not natural in this setting (even if, admittedly, it is standard in RL certainly for other reasons) and this paper does not exhibit any lower bound (except when there are no budget constraint, but in that case the results where already in the literature).

Finally, the writing of the paper lacks clarity and rigor. The function V(.,.) is not properly defined and it takes an infinite amount of time to decipher what the authors meant to say.

My suggestion is simple: do spend time polishing this paper, providing a better comparison with the literature, getting relevant lower-bounds (not just for B=T) and then this paper will be a really great paper. For now, it does not reach the high quality bar of ICLR.